# Mesoporous Silicas Obtained by Time-Controlled Co-Condensation: A Strategy for Tuning Structure and Sorption Properties

**DOI:** 10.3390/nano13142065

**Published:** 2023-07-13

**Authors:** Mariusz Barczak, Dorota Pietras-Ożga, Moaaz K. Seliem, Giacomo de Falco, Dimitrios A. Giannakoudakis, Konstantinos Triantafyllidis

**Affiliations:** 1Institute of Chemical Sciences, Faculty of Chemistry, Maria Curie-Sklodowska University, Maria Curie-Sklodowska Sq. 3, 20-031 Lublin, Poland; 2Department of Epizootiology and Clinic of Infectious Diseases, Faculty of Veterinary Medicine, University of Life Sciences of Lublin, 20-612 Lublin, Poland; 3Faculty of Earth Science, Beni-Suef University, Beni Suef 2722165, Beni Suef Governorate, Egypt; 4New Jersey Department of Environmental Protection, Trenton, NJ 08625, USA; giacomo.defalco@dep.nj.gov; 5Department of Chemistry, Aristotle University of Thessaloniki, 54124 Thessaloniki, Greece; dagchem@gmail.com (D.A.G.);

**Keywords:** mesoporous silicas, SBA-15, adsorption, diclofenac, co-condensation, functionalization

## Abstract

Mesoporous silicas synthesized by the co-condensation of two and three different silica monomers were synthesized by varying the time intervals between the addition of individual monomers, while the total time interval was kept constant. This resulted in different structural properties of the final silicas, particularly in their porosity and local ordering. One of the obtained samples exhibited an unusual isotherm with two hysteresis loops and its total pore volume was as high as 2.2 cm^3^/g. In addition, to be thoroughly characterized by a wide range of instrumental techniques, the obtained materials were also employed as the adsorbents and release platforms of a diclofenac sodium (DICL; used here as a model drug). In the case of DICL adsorption and release, differences between the samples were also revealed, which confirms the fact that time control of a monomer addition can be successfully used to fine-tune the properties of organo-silica materials.

## 1. Introduction

Ordered mesoporous silicas (OMS) have proved to be effective sorbents for the removal of many classical and emerging pollutants, such as pharmaceuticals, heavy metals, dyes and others [1,2,3,4,5,6,7,8,9,10]. This widespread interest has arisen as a result of several attractive properties of OMSs, such as high surface areas, tunable pore sizes with sharp distributions, and hydrolytically-stable structures, particularly at low pH values—among them, SBA-15 materials have become the most popular ordered silicas [11,12,13]. In addition, chemical modification of the hydroxylated silica surface with many commercially accessible monomers makes those materials particularly useful for environmental and biomedical applications, where a precise surface design plays a pivotal role [12,13,14,15,16,17]. In the case where the chemical moieties incorporated on the course of functionalization are terminal groups, namely, ≡Si-R, the resulting materials are called ordered mesoporous organosilicas (OMO). There is also a possibility to incorporate bridging groups, namely, ≡Si-R-Si≡; the resulting materials are then called periodic mesoporous silicas (PMS) [18,19,20].

There are two strategies for the surface functionalization of mesoporous silicas: (i) post-synthesis modification (grafting) and (ii) “on-course” functionalization (co-condensation) [21,22,23,24]. These two strategies lead to materials with diverse structures in the case of pore sizes and the distribution of organic groups. During post-synthetic grafting, the template-free material is treated with a modifying agent (usually a silica monomer) that reacts chemically with surface hydroxyl groups. The main drawback of this method is the insufficient control over the functionalization homogeneity, i.e., the grafted groups are not always evenly distributed. Usually, the density of the grafted functional groups is higher near the pore openings because they are more exposed to modifying monomers. This slows down the diffusion of further grafting molecules even more, contributing to even more inhomogeneity of the final materials.

The other functionalization strategy, namely, co-condensation, enables the direct one-pot incorporation of targeted functional groups within the silica mesoporous framework. A functional silica monomer is co-condensed with the silica precursor (or more of them) on the course of the synthesis, so that after the sol-gel process, both monomers are evenly distributed within the sample, and together form the resulting silica network. Compared with post-synthesis grafting, the co-condensation route synthesis is simpler and more advantageous because the formation of the mesoporous structure and surface-functionalization are achieved simultaneously in a one-pot synthesis. Organic groups are usually uniformly distributed within a polysiloxane framework without clogging the pores and their amount can be relatively precisely controlled; however, doping with a higher concentration may result in slower condensation rates and, in extreme cases, results in the formation of a disordered material. In particular, organosilanes with basic functionalities (e.g., amine and pyridine) can significantly disrupt the formation of an ordered mesoporous structure by over-catalyzing the sol-gel condensation step and additionally increasing the local pH, thus contributing to the destruction of the template micelles, which are stable in highly-acidic environments. As a consequence, the ordered structure and porosity deteriorate [25,26,27]. In most works, co-condensation is carried out with the use of two monomers: a structure-forming monomer (usually tetraethoxysilane, TEOS) and a functional monomer (e.g., aminopropyltriethoxysilane, APTS, bearing amine functional groups). 

Amine-functionalized nanoporous silicas are widely tested as sorbents of heavy metals, dyes and pharmaceuticals due to enhanced favorable interactions, both specific (e.g., the formation of hydrogen bonds) and non-specific (e.g., electrostatic interactions) between the drug molecules and an aminated surface [28,29,30,31,32]. In the case of amino groups, co-condensation usually turns out to be a much more effective way to obtain materials with a high concentration of these groups. For example, it was reported that the Cr(VI) adsorption capacity was almost ten times higher for silicas prepared by co-condensation than for those prepared by post-grafting [33].

One way to reconcile the high concentrations of amine functional groups and to still maintain an ordered structure is to extend the time between the addition of both co-condensing monomers. As an example, it was reported that extending the time between the addition of tetraethoxysilane (TEOS) and 3-aminopropyltriethoxysilane (APTS) resulted in an ordered mesoporous structure while the concentration of the amine groups remained high enough [34,35,36,37].

Herein, we propose a new strategy leading to the obtainment of ordered silica materials containing a high concentration of amine groups. It is based on adding a third “intermediate” monomer: 1,4-bis(triethoxysilyl)benzene (BTSB). Reports on the use of co-condensation of three monomers are very rare and come from our research group for both ordered [38,39] and amorphous organosilicas [40]; however, so far, they have only contributed to the observation of various physicochemical properties. In this work, for the first time in the literature, we investigate how the addition of the third monomer (BTSB) and the time at which it is added affects the physicochemical properties (e.g., porosity, local ordering, and surface chemistry) and the efficiency of the adsorption and release of the model drug—diclofenac sodium (DICL), chosen as a model pollutant. It is hypothesized that the addition of a BTSB monomer may significantly alter the final properties of the resulting materials, and in consequence, the resulting materials may have different sorption properties. To verify this hypothesis, a series of mesoporous amine-functionalized materials was synthesized via the one-pot co-condensation of two or three of the above-mentioned monomers.

## 2. Materials and Methods

### 2.1. Reagents

The following reagents were used as received: tetraethoxysilane (TEOS, 99%, Sigma-Aldrich, St. Louis, MO, USA), 3-aminopropyltriethoxysilane (APTS, 98%, Fluorochem, Glossop, UK), 1,4-bis(triethoxysilyl)benzene (BTSB, 96%, Sigma-Aldrich, USA), Pluronic P123 (P127, Sigma-Aldrich, USA), HCl (36%, POCH, Gliwice, Poland), NaOH (POCH, Poland), ethanol (EtOH, 99.8%, POCH, Poland), diclofenac sodium salt (DICL, >98%, Sigma-Aldrich, USA), and phosphate-buffered saline tablets (PBS, Life Technologies Ltd., Carlsbad, CA, USA). All chemicals were used as received, without further purification.

### 2.2. Synthesis of the Materials

The sample synthesis followed the SBA-15 synthesis protocol described previously [4,39,41]. The only difference was that for some samples, two monomers were used for the synthesis, and for others, three monomers. Briefly, 2 g of P123 was dissolved in 72 mL of 1.75 M HCl under stirring at 40 °C. After stirring overnight, the individual monomers were added in the following sequence: sample TA: 18 mmol TEOS, and after 90 min, 2 mmol APTS; sample BA: 9 mmol BTSB, and after 90 min, 2 mmol APTS; sample TBA1: 16 mmol TEOS, after 15 min, 1 mmol BTSB, and after 75 min, 2 mmol APTS; sample TBA2: 16 mmol TEOS, after 45 min, 1 mmol BTSB, and after 45 min, 2 mmol APTS; sample TBA3: 16 mmol TEOS, after 75 min, 1 mmol BTSB, and after 15 min, 2 mmol APTS. The resulting mixture was stirred at 40 °C for 24 h and aged at 100 °C for the next 24 h. The precipitated solid was washed with deionized water, filtered and dried at 70 °C. The template was removed by a triple extraction with acidified absolute ethanol (where each portion was composed of 147 mL of 99.8% ethanol and 3 mL of conc. HCl) at 70 °C. Then, the powders were washed with 500 mL of deionized water, filtered and dried at 70 °C.

### 2.3. Instrumental Characterization

The nitrogen isotherms were obtained at −196 °C by a Quantachrome 1200e analyzer. Before testing, the samples were degassed overnight at 110 °C in a vacuum. The BET-specific surface areas (S_BET_) were evaluated in the range of relative pressures *p*/*p*_o_ = 0.05–0.20. The total pore volumes (V_t_) were calculated by converting the amount adsorbed at *p*/*p*_o_ ~0.99 to the volume of the liquid adsorbate. The micropore volumes (V_m_) were calculated using the Saito and Foley (SF) method [42]. The pore size distributions (PSD) were calculated by the NLDFT method using the NovaWin software (Quantachrome, Boynton Beach, FL, USA). TEM and SEM microphotographs of randomly selected parts of the surface were collected using a Tecnai G20 X-Twin (FEI, Hillsboro, OR, USA) and Quanta 3DFEG (FEI, USA) microscope, respectively. The CHN elemental analysis was carried out using the CHN 2400 analyzer (Perkin Elmer, Waltham, MA, USA). Powder X-ray diffraction (XRD) patterns were recorded by using an Empyrean diffractometer (PANalytical, Malvern, UK) with a 0.02° size step and a 10 s time step covering a range of 0.5° < 2θ < 5.0°.

### 2.4. Adsorption and Release of Diclofenac

In each adsorption experiment, about 10 mg of adsorbent was shaken for 24 h min with 30 mL of a DICL solution with a known concentration. In the case of the kinetics measurements, the adsorption tests lasted between 2 min and 24 h. The equilibrium adsorption amounts were calculated by a mass balance using the following equation: a = (c_i_ − c_j_)·V·m^−1^, where c_i_ is the initial concentration (mg L^−1^), c_j_—the final concentration (mg L^−1^), V—the volume of the solution (L) and m—the mass of the adsorbent (g). Measurements of the DICL concentrations were carried out using the UV-VIS spectrometer Specord 200 (Analytic, Jena, Germany) at wavelength 278 nm after a previous filtration of the solution using 0.45 µm syringe filters. Release of the DICL was accomplished by loading the silica samples with DICL and then immersing them in 50 mL of unbuffered PBS solution and measuring the absorbance at 278 nm due to the DICL release.

## 3. Results

The sol-gel polycondensation scheme used to obtain the amine-functionalized mesoporous silica is shown in Figure 1. We applied a classical co-condensation scheme to obtain the amine-functionalized SBA-15 silica, i.e., the co-condensation of TEOS and APTS to obtain the sample TA. Analogously, the co-condensation of the BTSB and APTS led to the amine-functionalized sample BA. To keep the same Si/N molar ratio (90–10%) in the case of the DM1, 18 mmol of TEOS were co-condensed with 2 mmol of APTS, while in the case of the bissilylated BTSB monomer, 9 mmol of BTSB were used (because there are two silicon atoms in one BTSB molecule; therefore, we used half of the nominal number of mmols). The three remaining samples (i.e., the TBA1, TBA2, and TBA3) were synthesized by the co-condensation of the same three monomers: TEOS (16 mmol), BTSB (1 mmol) and APTS (4 mmol); however, the addition time of the BTSB was different each time, i.e., BTSB was added in the 15th (TBA1), 45th (TBA2) and 75th minute (TBA3) after the addition of the TEOS monomer (cf. Figure 1). 

To monitor the morphological changes resulting from the applied synthesis protocols, SEM and TEM analyses were run for all the samples. The SEM microphotographs are presented in Figure 2, and show that most of the obtained materials had a typical morphology of the SBA-15 structure. i.e., they were composed of “sausage-like” hexagonal motifs [38,43,44]. Of all the samples, only sample BA showed a completely different type of morphology: the hexagonal motifs were clearly shorter, and their mutual orientation was of a different type. This is obviously related to the fact that the dominant monomer in this sample was the BTSB and not the TEOS. The TEM images presented in Figure 3 reveal a well-ordered structure of all the samples, albeit the sample BA has a clearly visible hexagonally-oriented uniform mesopores. In the case of the BA sample, instead of domains of parallelly-oriented mesopores, short nanotubes chaotically-oriented to each other can be seen.

The XRD diffractograms of the samples are given in Figure 4a. In the case of the samples TBA1, TBA2 and TBA3, three well-resolved diffraction peaks can be observed in the range of ~0.9–1.8°. These peaks can be indexed according to the hexagonal p6m symmetry, indicating an ordered SBA-15 mesostructure: one sharp peak at ~0.9° indexed as (100) and two minor but distinct peaks at ~1.4° and ~1.7°, indexed as (110) and (200), respectively. In the case of the sample TA, the change in the XRD pattern testifies to the different types of symmetry in this sample, which was manifested by the presence of two diffraction peaks at ~0.8° and ~1.0°. In the case of the sample BA, only a poor mesoscopic order was observed, which was manifested by a broad diffraction peak with a maximum at ~0.7°. This observation is consistent with the already discussed SEM and TEM data, indicating an amorphous-like arrangement. Similar XRD patterns were previously found for functionalized silicas with the similar ratios of monomers [39,45]. The main reason for this behavior is related to the unmatched hydrolysis rates between a functional co-monomer and tetraalkoxysilane. Usually, co-monomers with organosilanes with basic functionalities (e.g., amine and pyridine) strongly interfere with the co-assembly process driven by the electrostatic and hydrogen bond formation between those groups and the block co-polymer, as well as the silicate species. In our previous work, we noticed that a small addition of certain monomers (as an example, BSTB) can have a neutral or positive effect on the local/mesoscopic ordering of the final materials; also in this case, the BTSB was the bridged monomer, which even in the small amounts added during the synthesis, led to a final well-ordered SBA-15 structure [38].

The calculated lattice constants from Bragg equation for the samples TBA1, TBA2 and TBA3 are 11.5 nm, 11.2 nm and 11.7 nm, respectively, which means that the mesoscopic order did not depend too much on when the BTSB was added during the synthesis (i.e., 15, 45 or 75 min, cf. Figure 1). For the samples TA and BA, a determination of their lattice constants was not possible due to low mesoscopic ordering.

The porous structure of the obtained materials was characterized by nitrogen adsorption–desorption tests; the isotherm curves are presented in Figure 4b, and the porous structure parameters are listed in Table 1. All the isotherms were of the type IV according to the IUPAC classification [46]. A type IV isotherm is characteristic of the mesoporous structure. The most striking feature was the presence of a sharp capillary condensation step as well as hysteresis loops due to the presence of a uniform array of mesopores with the same diameter. This is consistent with the TEM and XRD data discussed earlier. Hysteresis loops of the three TBA samples were typical for the nanoporous SBA-15 structure; however, the loops for the TA and BA samples were distantly different. The former was wider and flatter starting at lower *p*/*p_o_* values. This testifies to significant changes in both the pore shapes and sizes, and the overall worsening of the mesopores’ uniformity, suggesting that some of the mesopores could have been clogged.

Sample BA had an unusual adsorption isotherm with two well-separated hysteresis loops: (i) a loop in the *p*/*p_o_* range of ~0.6–0.75, which, similarly to the other samples, can be attributed to the presence of hexagonally-arranged mesopores due to the presence of a sacrificial P123 template, and (ii) a loop in the range ~0.85–1.0, typical for non-rigid aggregates of particles but that are still networks consisting of macropores which are not completely filled with nitrogen [46,47]. Indeed, considering the nitrogen adsorption tests and the TEM data, a significant nitrogen uptake close to the *p*/*p_o_* = 1 for the BA can be attributed to the capillary condensation between loosely arranged particles.

The synthesized materials showed a high S_BET_ with values between 439 and 747 m^2^/g. Comparing the samples TBA1–3, there was a noticeable S_BET_ increase associated with a delay in the BTSB addition during synthesis: a 15 min delay (TBA1) resulted in a S_BET_ = 439 m^2^/g, a 45 min delay (TBA2) resulted in a S_BET_ = 563 m^2^/g, while a 75 min delay (TBA3) resulted in a S_BET_ = 747 m^2^/g. The last two S_BET_ values were unattainable for the TA and BA samples synthesized using the two-monomer co-condensation route.

Similar trends were observed for the total pore volumes (V_t_) for all the samples but the BA. There was also a noticeable V_t_ increase associated with a delay in the BTSB addition for the samples TBA1–3 (i.e., 0.61, 0.78, and 0.92 cm^3^/g for the samples TBA1, TBA2 and TBA3, respectively). For each of them, the V_t_ value was also higher than the for TA sample, which did not contain the BTSB monomer at all. Sample BA had a much better developed porosity than the rest of the samples, which was manifested by a very-unusually high total pore volume (2.21 cm^3^/g), when compared to similar synthesis protocols for all the silicas. There are some reports that periodic mesoporous silicas can be highly porous [48,49], but nevertheless, such a large pore volume has never been observed before. Only specific pore-expanding strategies (e.g., the use of swelling agents) may result in silica materials (e.g., mesocellular foams) with comparable or higher pore volumes [50,51].

Another interesting trend here was the gradual increase in the micropore volume, V_m_, for the samples TBA1–3 with a prolonged addition time of BTSB. The V_m_ values were 0.19, 0.24 and 0.32 cm^3^/g for the samples TBA1, TBA2 and TBA3, respectively. This could suggest that the late addition of the BTSB favored the formation of microporosity. The porous structure of the synthesized materials largely depended on the moment when the BTSB monomer was introduced into the reaction mixture, although its amount was relatively small. 

The pore size distributions (PSD) presented in Figure 4c confirm the mesoporous nature of the obtained materials, as the main peaks are in the mesopore region. The maxima of the peaks located in the region 6–20 nm are given in Table 1 as the d_mes_ and should be considered the average sizes of the mesoporous channels. The average size of the mesopore channels was the smallest for the TA (8.4 nm), and slightly bigger for the BA (9.8 nm), while for the samples obtained by the co-condensation of three monomers, the d_mes_ values were in the range of 11.1–12.7 nm. All the samples except for the BA also had a significant fraction of micropores and smaller mesopores, with the average size ~2 nm. Thus, time-controlled co-condensation can be considered as a great tool not only to ensure high local/mesoscopic ordering (cf. Figure 4a), but also to optimize the resulting pore structure, including the S_BET_, V_t_, V_m_ and d_mes_.

At this point, it is worth considering what the mechanism of a delayed addition of the third component (BTSB) on the resulting structure of the samples TBA1, TBA2 and TBA3 was. Most likely, there was no single decisive factor responsible for the observed differences and several aspects should be taken into account.

Firstly, the addition of BTSB after a shorter time interval (15 min) after the introduction of TEOS implies that the co-condensation of both monomers started when the initial “embryo” mesostructure was not fully formed [52]; thus, it was more prone to co-condensation even taking into account the unmatched rates of hydrolysis of the two monomers. A BTSB-delayed addition as long as 75 min resulted in the poor co-condensation of both monomers because a robust and well cross-linked structure was formed by the TEOS condensation, in contrast to the initial phase of synthesis, where the condensation rate was lower than the hydrolysis rate [53]. BTSB, therefore, could not be successfully built and evenly distributed into the silica framework. This fact is clearly confirmed by the shape of the hysteresis loop on the isotherm of the TBA3. This type of stretched hysteresis loop is characteristic of the presence of both open and blocked pores [54], which means that BTSB molecules are located at the entrances to the mesopores, creating plugs, causing a partial blockage of the entrances to the pores.

Secondly, it has been reported that the presence of BTSB may increase the interactions between the hydrophobic benzene bridges and the dehydrated PEO shell of the polymeric template, resulting in materials with a lower surface area and total pore volume than TEOS-condensed-only silicas [55]. A Comparison of specific surface areas and pore volumes of the bare (i.e., non-functionalized) SBA-15 silicas obtained in our previous works [45,56,57] with the TBA1, TBA2 and TBA3 silicas obtained in this work, showed that the S_BET_ and V_t_ values did indeed decrease; however, this effect gradually disappeared with an increasing delay in the BTSB addition, which in later phases of its addition rather tended to clog the pore openings (vide supra).

Finally, entropic effects may also play a role as has been recently shown [58]. These effects may arise from the incorporation of hydrophobic bridges into the silica framework, resulting in so-called “hydrophobic hydration”, i.e., the formation of hydrophobic domains inducing a local ordering of the interfacial water. These low entropy “hotspots” are able to influence the formation of a silica framework and, therefore, the final properties of organically-modified silicas, including an alteration of the surface chemistry [58]. 

The results obtained from a CHN elemental analysis allowed us to obtain interesting information about the efficiency of the co-condensation. As expected, the amount of carbon depended on the amount of organosilica monomers co-condensed with TEOS; therefore, the carbon content was the lowest for the sample BA (~5%), intermediate for the samples of TBA (~9–12%) and the highest for the sample BA (38%). The content of nitrogen in the final samples also varied with the highest value observed for the sample TA (1.52%) and the lowest for the sample BA (0.77%). Knowing the theoretical contents of these elements, we determined the effectiveness of the functionalization in relation to carbon and nitrogen, calculated as a ratio of the observed and theoretical contents of carbon and nitrogen (C_eff_ and N_eff_, respectively; cf. Table 1). For all the samples, the C_eff_ efficiencies were close to the theoretical values (91–102%); however, the N_eff_ efficiencies were, in the same time interval, significantly lower (52–71%), which means that part of the carbon may have come from the unremoved P123 template instead of the APTS, which apparently had not been fully co-condensed into the nascent siloxane framework. Assuming then, that similar amounts of the unremoved template remained in all the final samples, the functionalization efficiencies of the obtained samples were most probably lower than those suggested by the carbon content. On the other hand, taking into account some portion of the unremoved polymer template present in the final structures, the real nitrogen content may have been slightly higher.

The resulting silicas’ potential as environmental adsorbents was tested for the removal of a model drug: diclofenac sodium (DICL) from an aquatic environment. We have previously shown that the adsorption of diclofenac sodium slightly changes in the pH range of 5–8 when amine-functional silica was used as the sorbent [29]; therefore, in a similar manner of previous investigations, we carried out the adsorption experiments in unbuffered solutions (pH ≈ 5.5–6.0) [29,45,59]. Adsorption isotherms of the samples studied are given in Figure 5a, and the observed DICL uptakes (i.e., static sorption capacities, SSC) are given in Table 2. The adsorption equilibrium data were modeled following the Langmuir adsorption model and the obtained curves are presented in Figure 5. Table 2 lists the fitted parameters. The TA adsorbed the most DICL of all the samples (i.e., a SSC value of 251 mg/g), and the BA adsorbed noticeably less (206 mg/g), while the lowest amounts adsorbed were observed for the TBA samples and they were in the range of 114–153 mg/g. 

A comparison of the DICL uptakes with the physicochemical characteristics data provided valuable information on the factors influencing the adsorption efficiencies. The adsorbed amounts of DICL did not depend remarkably on the structural parameters (such as, for example, the porous structure or local ordering). No correlations could be found between the porous structure parameters (i.e., S_BET_, V_t_, V_m_, and d_mes_) and the DICL uptakes. On the other hand, the nitrogen content (corresponding to the number of amino groups) was clearly an important factor governing the DICL removal efficiency. This observation has been confirmed in dozens of publications, including our work explaining the mechanism of interaction of diclofenac with APTS amine groups [45], based on the formation of hydrogen bonds between a DICL carboxylate anion and a protonated amine group. 

An interesting observation was that the sample BA adsorbed much more DICL than expected considering the nitrogen content. In the case of the BA, the SSC value was lower only by ~20%, with a twice-lower content of amino groups compared to the TA sample. There may have been two main reasons explaining this relatively-high DICL uptake by the BA sample.

Firstly, the hydrophobicity of the BA sample was definitely higher than that of the other samples, which resulted from the abundance of benzene rings in its structure (i.e., the carbon content was 38%, cf. Table 1). It is well known that hydrophobic interactions are one of the three mechanisms responsible for DICL adsorption [45,47,60]. The other two are the electrostatic interactions between the silica surface and DICL and the aforementioned mechanism based on the formation of hydrogen bonds between DICL carboxylate anions and protonated amine groups [45,59]. Therefore, the presence of a large number of benzene bridges may have favored the adsorption of diclofenac due to hydrophobic interactions.

Secondly, analyzing the SEM and TEM images as well as the adsorption isotherm of the sample BA, a remarkably different morphology and porosity was observed. A greater fragmentation of this sample (cf. Figure 2 and Figure 3) as well as its hierarchical porosity (cf. Figure 4b,c) may have significantly increased the accessibility of amine groups to the DICL molecules; in particular, the location of the amine groups in larger pores and interparticle spaces can certainly have had a positive effect on the final DICL uptake. It seems that it is not the number of amino groups alone that determines the possibility of their interaction with diclofenac, but also their sufficient exposure. This is particularly important in the case of such relatively-large molecules as diclofenac, which may not be able to enter into micropores.

Studies on the release of diclofenac from the obtained materials were also performed and are presented in Figure 5b. Since we did not control how much DICL was adsorbed in the samples, the relative release profiles are shown in the graph. The presented profiles are quite similar in shape with a relatively fast release of DICL during the first two hours (a so-called burst effect [61,62]), gradually turning into a more sustainable release over time. Among all the samples, the TBA3 showed a slightly different release profile characterized by a more sustainable release of DICL with a remarkably less-pronounced burst effect. The reason for such a favorable DICL release from the TBA3 may have been its porous structure with a large share of micropores in the total porosity. This caused a portion of the DICL bound to the amino groups in the micropores to be more difficult to be separated from the surface, than the DICL loaded in the mesopores. This proves that through the time-controlled co-condensation proposed in our work, it is possible to tune (to some extent) not only the final properties of materials but also their drug release profiles. 

## 4. Conclusions

The morphology, porous structure, surface chemistry and hydrolytic stability turned out to be dependent not only on the type and amount of the silica monomers used for the synthesis but also on the time intervals between the addition of these monomers. The local ordering of the synthesized materials differed depending on the presence of a relatively small amount of BTSB added during the synthesis, while the porous structure depended on the time at which the BTSB was added. Due to the presence of surface-exposed amine groups, all the materials were characterized by high diclofenac uptakes (i.e., 114–251 mg/g). Differences in the uptakes were related mainly to the number of amine groups; however, in the case of the sample BA, a massive amount of benzene ring bridges apparently induced additional hydrophobic interactions with the DICL, significantly contributing to the overall uptake (206 mg/g), even despite the low content of amine groups. The three-monomer synthesis protocols, however, that changed the time interval between the addition of individual monomers (which is often a forgotten factor) can be used to regulate the porous structure and surface chemistry of the resulting silica materials and, consequently, the sorption and desorption properties.

## Figures and Tables

**Figure 1 nanomaterials-13-02065-f001:**
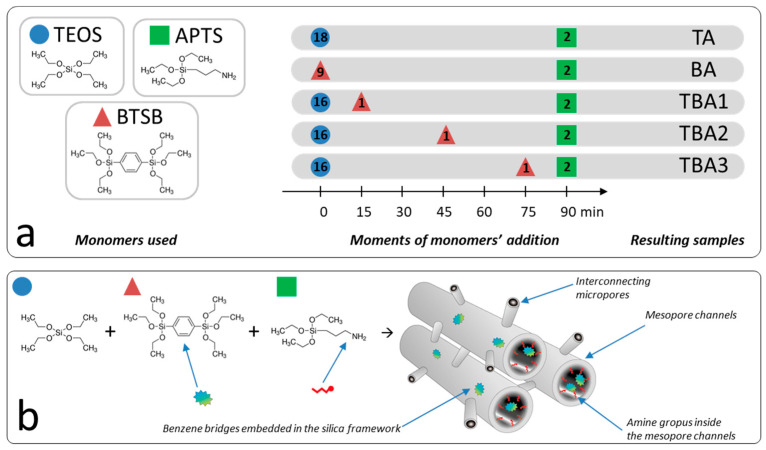
Scheme of the synthesis routes used in this study to obtain the resulting samples: synthesis protocol (the numbers inside the figures indicate the amount of added mmols for the relevant monomer) (**a**) and schematic presentation of the co-condensation reaction of three monomers (i.e., TEOS, BTSB, and APTS) used to obtain the TBA1, TBA2 and TBA3 mesoporous silicas (**b**).

**Figure 2 nanomaterials-13-02065-f002:**
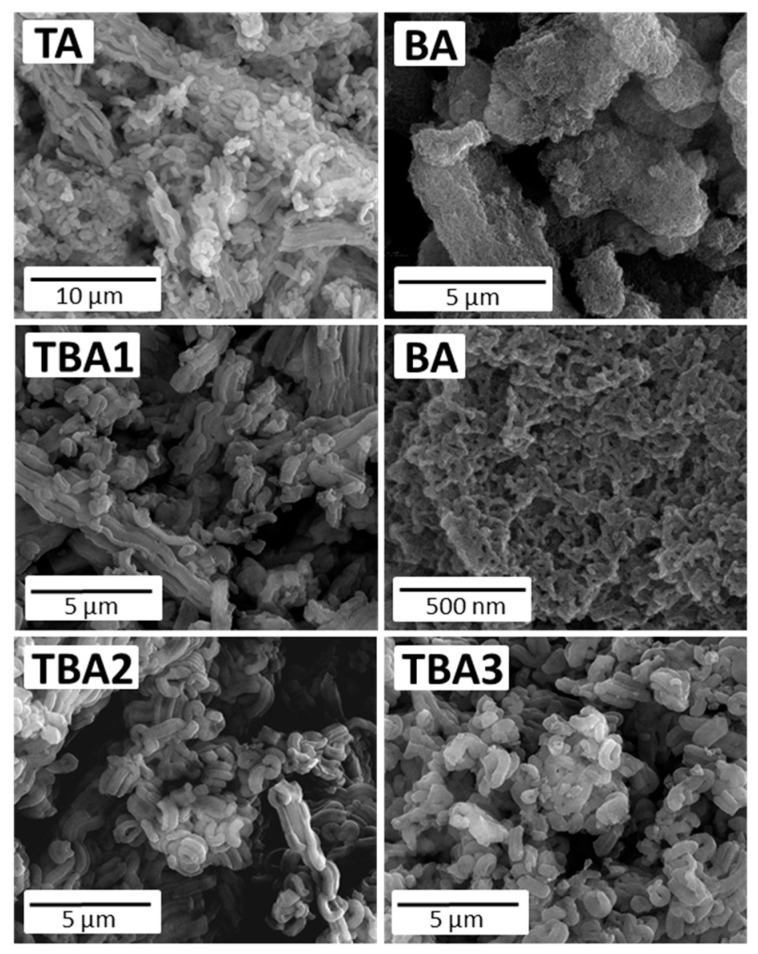
SEM images of the obtained mesoporous silicas.

**Figure 3 nanomaterials-13-02065-f003:**
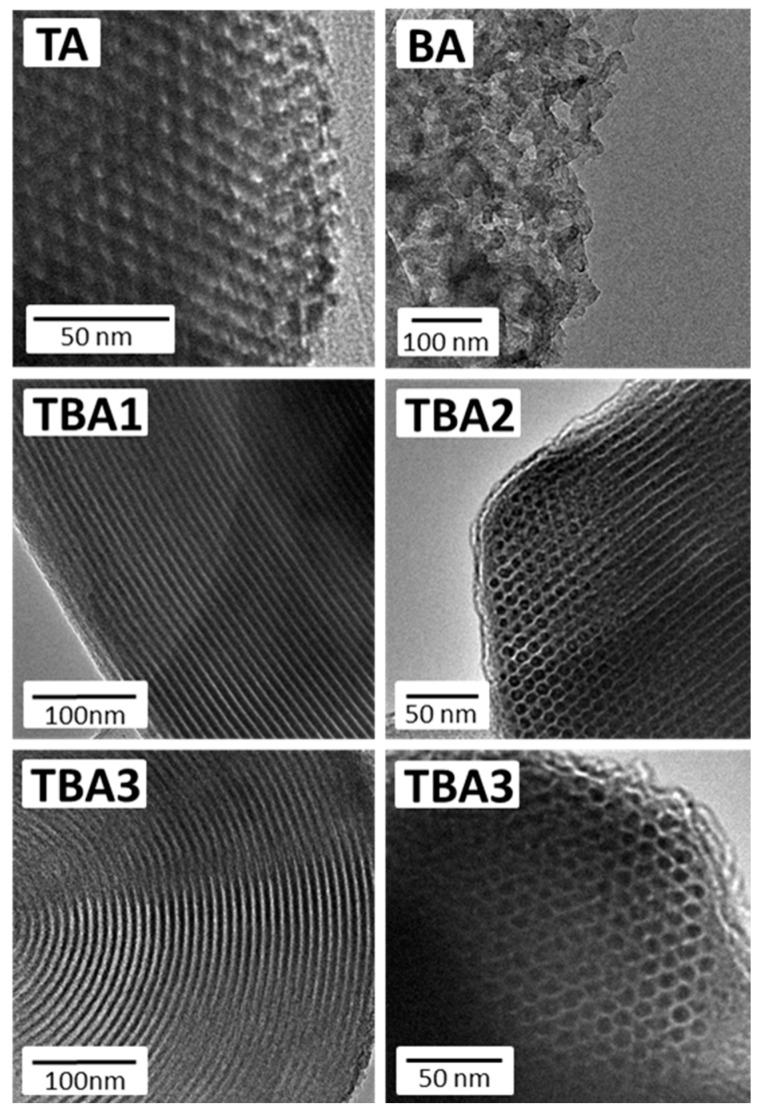
TEM images of the obtained mesoporous silicas.

**Figure 4 nanomaterials-13-02065-f004:**
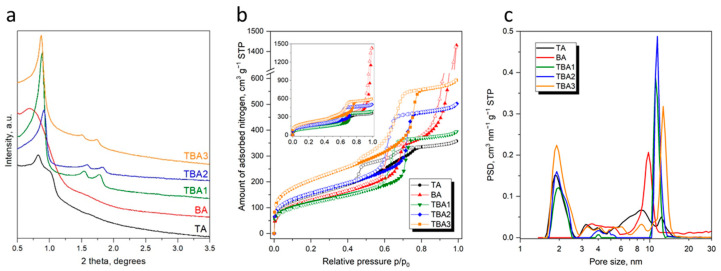
XRD low-angle diffractograms (**a**), nitrogen adsorption isotherms (**b**), and pore size distributions (**c**) for the studied silicas.

**Figure 5 nanomaterials-13-02065-f005:**
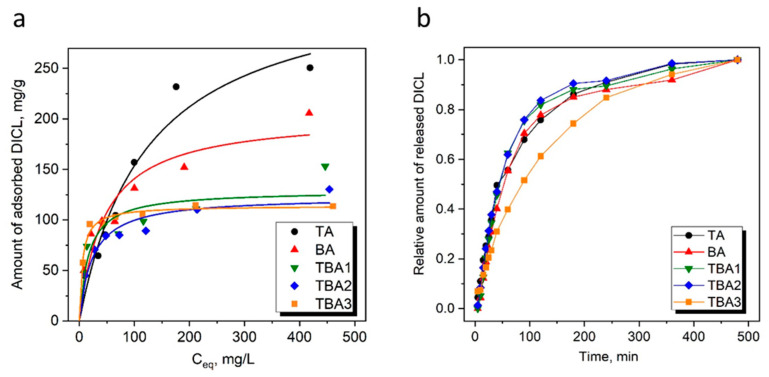
Adsorption isotherms and their fitting with the Langmuir model (**a**) and release profiles of DICL from the samples studied (**b**).

**Table 1 nanomaterials-13-02065-t001:** Selected structural and chemical properties of the studied OMSs.

Sample	CHN Elemental Analysis					Porous Structure Parameters			
	C (%)	H (%)	N (%)	C_eff_ (%)	N_eff_ (%)	S_BET_ (m^2^/g)	V_t_ (cm^3^/g)	V_m_ (cm^3^/g)	d_mes_ (nm)
TA	5.25	2.38	1.52	95	71	541	0.55	0.23	8.4
BA	38.03	4.55	0.77	97	51	470	2.21	0.20	9.8
TBA1	10.84	2.77	1.20	102	58	439	0.61	0.19	11.1
TBA2	9.94	2.48	1.07	94	52	563	0.78	0.24	11.4
TBA3	9.64	2.64	1.10	91	53	747	0.92	0.32	12.7

S_BET_: specific surface area by BET method, V_t_: total volume of the pores, V_m_: volume of micropores, and d_mes_: size of primary mesopores read from the maximum on the PSD curve.

**Table 2 nanomaterials-13-02065-t002:** Langmuir fitting parameters and the observed uptakes (SSC) of the studied OMSs.

Sample	Langmuir Fitting			SSC (g_DICL_/g)
	q_m_	K_L_	R^2^	
TA	341	0.008	0.95	251
BA	204	0.008	0.87	206
TBA1	129	0.028	0.74	153
TBA2	123	0.042	0.89	130
TBA3	114	0.031	0.94	114

## Data Availability

The data presented in this study are available on request from the corresponding author.

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
