# Peer review of "Mesoporous Silicas Obtained by Time-Controlled Co-Condensation: A Strategy for Tuning Structure and Sorption Properties"

_nanomaterials, 2023, doi:10.3390/nano13142065_

Round 1

Reviewer 1 Report

The paper presents an investigation on preparation of functionalized mesoporous silica sorbents using co-condenstation of TEOS and functionalized organosilicate precursors. The advantage of the co-condensation is the better control over the homogeneity of distribution of the functional groups on the pore surfaces. While it is a well-known method, the novelty of the present study is the use of a third silica precursos in a minor amount, which served as modifier of the materials morphology and pore surface properties.

The paper is well written and can be recommended for publication after improvements as described below.

1. The novelty of the use of a third silica precursor should be checked by literature search, and depending on the result, should be better emphasized in the Introduction.

2. The use of aminopropyl and other precursors in co-condesation has been demonstrated on ordered mesoporous silica in following recent studies, which can be mentioned in the instroduction. 10.3390/ma14030628 , 10.1016/j.ceramint.2018.11.152 , 10.1080/1536383X.2019.1593154

3. An explanation of the mechanism of delayed addition of the third component BTSB on the resulting structure should be suggested in the Discussion and in the Conclusions. So far, it is only the observed effect is reported.

4. Check occurrences of abbreviations of APTES and APTS

5. Check the name of the manufacturer and the model type of the diffractometer.

6. Subsection title Activation treatment does not correspond to its content, as there was no explicit activation procedure involved. More traditional title such as synthesis could be used instead, if authors agree.

3. lines 140-141 check the grammar of the sentence.

4. line 177 Citation is necessary here.

5. line 185 check grammar

6. line 217 - extra comma

Check grammar and punctuation

Reviewer 2 Report

The article is interesting. However, it contains some shortcomings.

Firstly, there is no scheme for the synthesis (co-condensation) of monomers into polymeric sorbent structures. Therefore, this scheme should be added and shown.

Secondly, no designations and no magnification were indicated for Figures 2 & 3. In addition, too many (30), mostly low-quality, images were shown. Therefore, the authors should select and show the three best-quality images of each Figure 2 and 3 at optimal magnification instead of multiple low-quality images.

Thirdly, the purpose of this study was the synthesis and study of mesoporous silica sorbents. However, the pore size in these sorbents was not determined and not reported. For example, Table 1 shows the specific surface value, and pore volumes, but the size of pores is absent. This omission must be rectified, namely, the pore size in the sorbents should be indicated to prove that the pores are indeed mesomorphous.

Fourthly, the method for determination of the volume of micropores (Vm) should be added to section 2.3.

Fifthly, why are there no calculations of interchain distances (d) in the sorbents using the Bragg equation if the 2Ï´ values of the peaks on X-ray patterns are known? For example, if 2Ï´ is 0.9°, 1.4°, and 1.7°, then it can be calculated that the d-value will be 9.8 nm, 6.3 nm, and 5.2 nm, respectively, corresponding to meso-capillaries or mesopores. In addition, the authors should clarify the term “ordering” of the samples.

Lines 214 & 230. Remark: What is Vp? This term before was not mentioned in this article and it must be explained. In addition, the method of its determination should be described.

Table 1. Sample TBA1. Remark: Since the theoretical efficiency of polycondensation cannot exceed 100%, the relative carbon addition to the sorbent due to polycondensation cannot be 112%. The authors should recalculate this result.

Line 244. ...(the efficiency) for sample BA it is noticeably higher than 100%,  Remark: Table 1 does not show that the relative efficiency of C or N for BA sample is higher than 100%, namely, it is  97% for C and 51% for N. The authors should correct this statement.

Line 249. ... (the N efficiency) for sample BA reaching a value higher than 70%. Remark: Table 1 does not show that the relative efficiency of N for BA sample is higher than 70%, namely, it is 51% only.  The authors should correct this statement.

Line 268-269. Adsorbed amounts of DICL do not depend on the porous structure or ordering of the sorbent... Remark: This statement is incorrect - vice versa, the higher is ordering of sorbent the lower should be sorption value, SSC. In addition, the authors should clarify the term “ordering”

Lines 270-271. On the other hand, the nitrogen content (corresponding to the amount of amino groups) is clearly an important factor governing the DICL removal efficiency. ... Remark: This statement is incorrect - for example, the N content in BA sample is the lowest (0.77%), but DICL sorption by this sample is higher than by TBA1, TBA2 or TBA3 samples despite that their N content is higher (1.07-1.20%) that in BA sample (0.77%). Instead, the authors should offer another explanation, such as that written in lines 281-293.

Line 27. Remark: Put article “the” before “removal”.

Line 32. Remark: Put article “the” before “most”.

Line 35. Remark: (1). Remove article “the” before “precise”; (2). The verb’s singular form, “plays", should be used here instead of the plural form, "play".

Line 40. Remark: Replace “to” with “for” the surface....

Line 43. Remark: Put a comma after “grafting”

Line 52. Remark: Put article “the” before “direct”.

Line 54. Remark: Replace “on” with “in” the course...

Line 64. Remark: Write the word "supercatalysing" with a hyphen, "super-catalyzing".

Line 76. Remark: The verb’s singular form, “remains", should be used here instead of the plural form, "remain".

Line 79. Remark: Put article “a” before “relatively”

Line 94. 2.2. Activation treatments. Remark: Since this section describes the synthesis of silica sorbents, the title of the section should be changed to “2.2. Synthesis of silica sorbents”.

Line 104. Remark: Put article “the” before “next”

Line 111. Remark: Put a hyphen between BET & specific, “BET-specific”...

Line 114. Remark: Put article “the” before “liquid”

Line 115. Remark: Put article “the” before “NLDFT”

Line 137. Remark: Put article “the” before “co-condensation

Line 139. Remark: Put a hyphen: “bis-silylated”

Line 140. Remark: (1). Remove the word “only” before 9 mmol; (2). Write “because there are two silicon atoms in one”...

Line 156. Remark: Write “a different” instead of “adifferent”

Line 158. Remark: Put a hyphen: “well-ordered”

Line 175. Remark: Put article “an” before “amorphous-like”

Line 177. Remark: Remove article “the” before “similar”

Line 185. Remark: Replace article “an” with “a” before “final”

Line 195. Remark: Put article “the” before “overall”

Line 214. Remark: Remove “the” before “for”

Line 223. Remark: Write “time” instead of “timet

Line 228. ...considered as a great tool...  Remark: Remove “as” in this phrase

Line 240. Remark: Put article “the” before “knowing”

Line 255. The resulting silicas’ potential as environmental adsorbent were tested for the removal... Remark: The English should be corrected here, namely, (1) put article “an” before “environmental” and (2) “write “was” instead of “were”  

Line 262. Remark: Put article “the” before “Langmuir”

Line 299. tTBA3. Remark: Write TBA 3 instead.

Round 2

Reviewer 1 Report

In the revised version, authors improved the discussion of their results by interesting considerations. The paper is recommended for publication in Nanoaterials, after a careful check of the text. A few omissions noticed are listed below.

line 272, check spelling

line 320, sentence not finished.

ref 1, authors list is incorrect

ref 14, authors list is incorrect

ref 17, doi number is incorrect

ref 19, journal volume is incorrect

ref 25, authors list is incorrect, and doi is given twice.

ref 26, authors list is incorrect

ref 30, doi is given in wrong format

ref 34, citation format is not according to the journal style

ref 36, doi number is incorrect

ref 45, journal name abbreviation is incorrect, Glass should be not abbreviated

ref 47, journal name, volume and page numbers are missing

ref 59, mistake in paper title

ref 61, doi number is incorrect

ref 62, citation style is wrong

Author Response

We would like to thank to the Reviewer for so detailed insight - we of course corrected all the indicated errors and we recheck the manuscript.

Reviewer 2 Report

Since the authors revised the manuscript according to the remarks of the reviewer, I can recommend the revised paper for publication 

No comments

Author Response

We would like to thank the Reviewer for such detailed insight, which definitely contributed to the quality of the final manuscript.